# Improving Blast Performance of Reinforced Concrete Panels Using Sacrificial Cladding with Hybrid-Multi Cell Tubes

**Mahmoud Abada, Ahmed Ibrahim * and S.J. Jung**

Department of Civil and Environmental Engineering, University of Idaho, 875 Perimeter Dr. MS 1022, Moscow, ID 83844, USA; moha4943@vandals.uidaho.edu (M.A.); sjung@uidaho.edu (S.J.J.)
* Correspondence: aibrahim@uidaho.edu

**Abstract:** The utilization of sacrificial layers to strengthen civilian structures against terrorist attacks is of great interest to engineering experts in structural retrofitting. The sacrificial cladding structures are designed to be attached to the façade of structures to absorb the impact of the explosion through the facing plate and the core layer progressive plastic deformation. Therefore, blast load striking the non-sacrificial structure could be attenuated. The idea of this study is to construct a sacrificial cladding structure from multicellular hybrid tubes to protect the prominent bearing members of civil engineering structures from blast hazard. The hybrid multi-cell tubes utilized in this study were out of staking composite layers (CFRP) around thin-walled tubes; single, double, and quadruple (AL) thin-walled tubes formed a hybrid single cell tube (H-SCT), a hybrid double cell tube (H-DCT), and a hybrid quadruple cell tube (H-QCT). An unprotected reinforced concrete (RC) panel under the impact of close-range free air blast detonation was selected to highlight the effectiveness of fortifying structural elements with sacrificial cladding layers. To investigate the proposed problem, Eulerian–Lagrangian coupled analyses were conducted using the explicit finite element program (*Autodyn/ANSYS*). The numerical models' accuracy was validated with available blast testing data reported in the literature. Numerical simulations showed a decent agreement with the field blast test. The proposed cladding structures with different core topologies were applied to the unprotected RC slabs as an effective technique for blast loading mitigation. Mid-span deflection and damage patterns of the RC panels were used to evaluate the blast behavior of the structures. Cladding structure achieved a desired protection for the RC panel as the mid-span deflection decreased by 62%, 78%, and 87% for H-SCT, H-DCT, and H-QCT cores, respectively, compared to the unprotected panels. Additionally, the influence of the skin plate thickness on the behavior of the cladding structure was investigated.

**Keywords:** blast loading; sacrificial cladding structure; hybrid-multi cell tubes; hydrocodes; Coupled Eulerian; Lagrangian algorithm



## 1. Introduction

The investigation of buildings capability and their structural elements to tolerate explosions has become an active area of research in structural engineering fields. Blast loads do not only result from terrorist attacks but are also due to industrial or transportation-related accidents that may have flammable materials such as petroleum, propane, etc. These accidents destroy these buildings and significant human casualties of their occupants. Thus, protecting structures against blast hazards become inevitable for military and civilian governments in many countries.

Typically, common structures are not designed to endure extreme load conditions such as blast loading, so it become inevitable for many governmental agencies to design and retrofit structures against blast loads. A few of the available solutions for shielding structures against blast risks are as follows: (a) Attaching a thick concrete covering to steel members. However, this approach has many flaws such as a large deadweight added to

the structure. Furthermore, explosion tests showed that concrete completely pulverizes and causes casualties due to flying fragmentations [1]. (b) Coating the internal walls of the structure by LINE-X or POLYUREA but this technique is costly for application in the construction industry. (c) Another alternative is glass laminated aluminum reinforced epoxy (GLARE), which is a metallic sheet consisting quite a lot of very thin layers of aluminum interspersed with layers of prepreg glass-fibers, bonded with a matrix (epoxy). It is an excellent resistance against impact, blast loading, and fire [2]. However, this material is costly for application in the construction industry. (d) Additionally, nanomaterials acquire an important contribution in the blast mitigation of concrete structures. However, producing nanomaterial is a costly process, and the quantities required for the construction industry are massive.

Recently, sacrificial cladding structures have attracted more attention as effective blast alleviation techniques due to their superior energy absorption capabilities and low cost [3–5]. The sacrificial cladding structure has two layers (an outer plate and an inner core layer) as illustrated in Figure 1a. The face plate's function is to evenly distribute the blast pressure across the crushable core layer which gradually deforms and absorbs large amount of impact energy, thus the pressure transferred to the structure is attenuated. The working concept of sacrificial cladding structure is when an explosive detonates in the proximity of infrastructures, it undergoes an instantaneous impulsive load for a very short time interval. To safeguard structures from these devasting impulses, the proposed cladding structures must be installed at the façade of those structures as shown in Figure 1a. During the explosion, the sacrificial cladding layers will be exposed to a high load ($P_0$) pulse of short duration. The objective of the entire sacrificial cladding structure is to convert a high impulsive load with short duration to lower load with long duration as illustrated in Figure 1b. In order to evade permanent damage to the unprotected structure (main structure) from taking place, the failure load of the sacrificial cladding structure ($F_{max}$) must be maintained less than the minimum elastic capacity of the main structure. Thus, the applied load conveyed to the main structure is minimized [6,7]. The efficiency of the sacrificial cladding structures mainly depends on the amount of energy absorbed by its core [8]. TW structures with various materials and shapes have been employed as effective energy absorber components in crashworthiness applications. It can attenuate a large portion of impact energy by converting it into plastic energy when it is deformed by the applied pressure produced by the shock wave [9]. Hence, they can be exploited as an effective core layer for sacrificial cladding structures.

Several studies have been performed to understand and evaluate the blast behavior of protected structural elements under the impact of blast loads. Hanssen et al. [10] used aluminum foam (ALF) with different densities as sacrificial layers. They realized that the transmitted energy and impulse varies according to the foam density. Mazek et al. [11] experimentally and numerically investigated the performance of the RC panels with and without aluminum foam (ALF) layers and rigid polyurethane foam (RPF) layers to fortify the RC panels subjected to blast loads. The blast performance of the RC panel was improved by 45% and 70% for the panels strengthened by RPF and ALF layers, respectively with respect to the bare RC panel. Van et al. [7] executed a detailed numerical and experimental study on the blast performance of a sacrificial cladding with various configurations of composite tubes. They found that using sacrificial cladding maintains structural integrity. Codina et al. [12] introduced a novel sacrificial cladding for minimizing blast damage of RC columns by covering the structural element with reinforced resin panels. The experimental findings implied a reduction in the final deflection compared to the unprotected column of 57.4% for the steel jacket and 66% for the reinforced resin panels. Al-Rifaie et al. [13] numerically investigated the blast performance of novel sandwich panels with unconnected graded layers. They found that sandwich structures with graded layers had a superior blast performance compared to the ones with ungraded core layers.

This paper presents sacrificial cladding structures with a core layer consisting of three different groups of hybrid multi-cell tubes. The new proposed sacrificial cladding

structures were mounted in front of the RC concrete panel in order to enhance the blast behavior of the RC panel. An unprotected RC panel exposed to close-range free air blast test executed by Wang et al. [14] was selected to be a control panel to highlight the impact of the new proposed mitigation techniques.

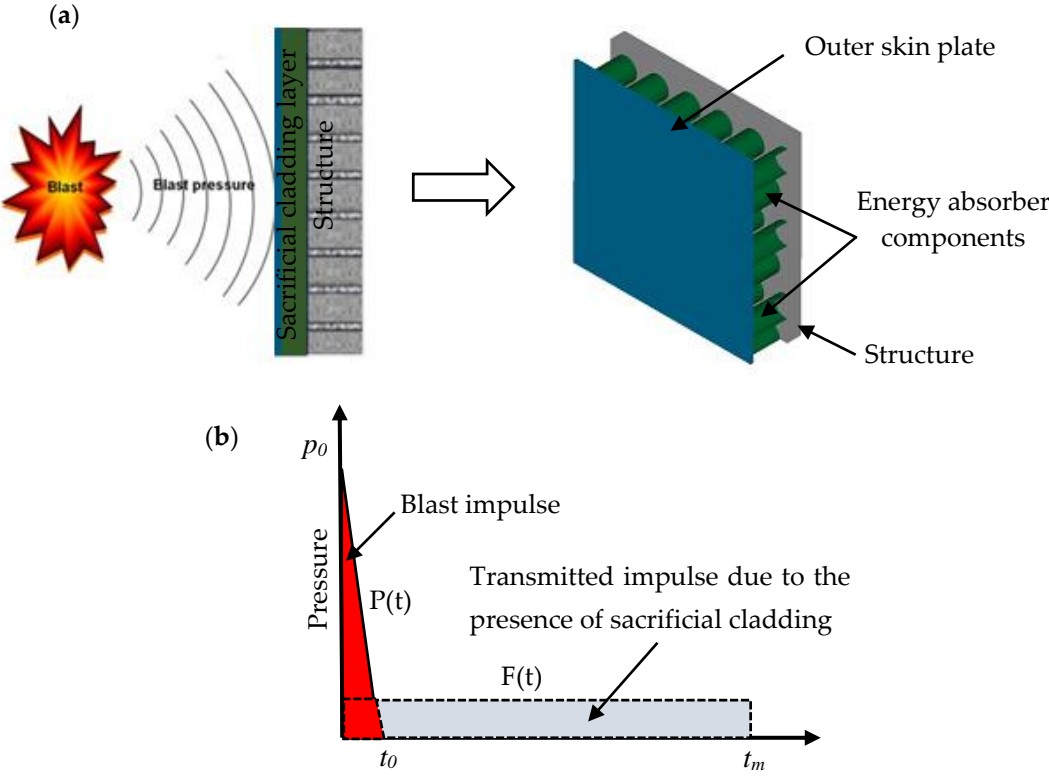

**Figure 1.** (**a**) Schematic of sacrificial cladding layers, (**b**) sacrificial cladding structure concept.

## 2. Numerical Model Validation

Recently, numerical modeling has been replaced blast field tests in the predesign stage of structures as well as investigating their performance under extreme load conditions. Field blast test is a reliable approach to examine the dynamic performance of blast resistant structures. However, the blast field test is expensive to conduct in every site and sometimes it is impossible to carry out such field tests because of their size, complexity, safety, and cost [15]. In this study, the numerical approach was adopted to examine the dynamic performance and the failure of the introduced structures under the effect of blast loading. All numerical models were carried out by utilizing the commercial explicit finite element program *Autodyn/ANSYS 2019 Version R2*. Numerical results obtained by the FEA were verified by the data obtained from the field blast test executed by Wang et al. [14]. Reliable numerical model validated against measured field data is an effective tool to analyze the structural performance under blast impact. Three one-way square RC slabs subjected to close-in blast loadings with different scaled distances (*Z*) of 0.518 and 0.591 $m/kg^{1/3}$ were considered in this study. The dimensions of the panels are shown in Table 1. The setup of the field blast test is shown in Figure 2a. The specimens' reinforcements were constructed using 6 mm steel bars in both directions with spacing of 75 mm in-between bars as shown in Figure 2b. The reinforcement ratio in both directions was 1.43%. The concrete compressive strength was 39.5 MPa; tensile strength of 4.2 MPa; and Young's modulus of 28.3 GPa. The reinforcement has a yield strength of 600 MPa and Young's modulus of 200 GPa. The test rig was constructed, as illustrated in Figure 2. The RC slabs were fixed on the sides to avoid lifting during testing [14]. TNT explosive charges were hanged above the center point of the specimens at a specific stand-off distance (*SoD*) (300, 400,

and 500 mm) measured from the center of the explosives to the upper surface of the slab. Table 1 summarizes the experimental program.

**Table 1.** Blast field test data [14].

| Panel | Dimensions (mm) | Charge Weight (gm) | *SoD* (mm) | Z (m/kg1/3) |
|---|---|---|---|---|
| Panel A | 750 × 750 × 30 | 130 | 300 | 0.591 |
| Panel B | 750 × 750 × 30 | 190 | 300 | 0.518 |
| Panel C | 1000 × 1000 × 40 | 310 | 400 | 0.591 |
| Panel D | 1000 × 1000 × 40 | 460 | 400 | 0.518 |

(**a**)  (**b**)

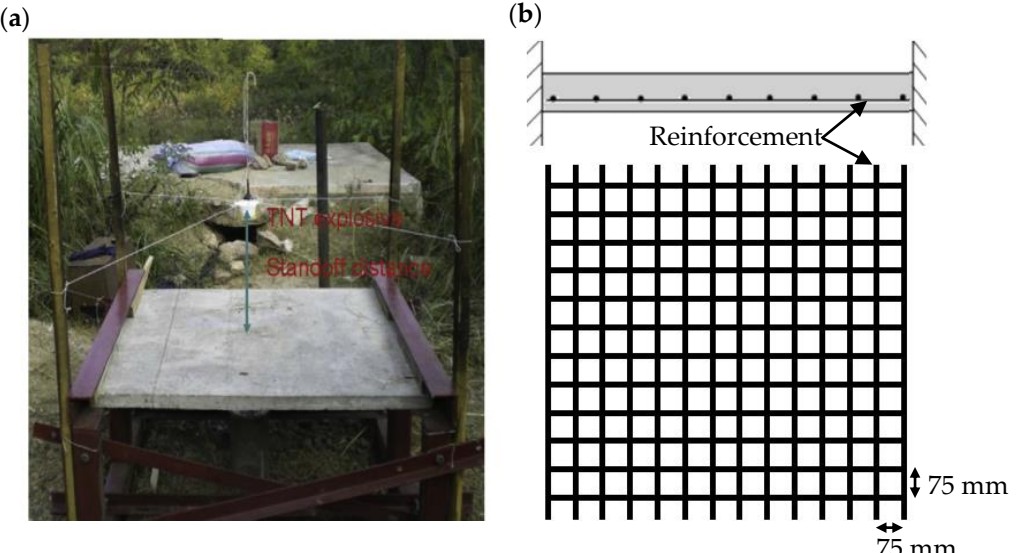

**Figure 2.** (**a**) Setup of the experimental blast test [14], (**b**) Slab cross-section and reinforcement.

*2.1. Numerical Modeling*

2.1.1. Material Models

Concrete Model

Dynamic modelling for concrete structures exposed to blast load still a challenge and needs high fidelity computer simulations. It is important to have a precise finite element model that represents concrete material characteristics with respect to high-stress rate effect. In this study, Riedel–Hiermaire–Toma (RHT) model was utilized to simulate the hydrodynamic behavior and crack trajectory of the concrete under the impulsive blast load. RHT model is a complex plasticity model established by Riedel et al. [16] and adopted for brittle materials (concrete). This model considers numerous features for instance; strain hardening, pressure hardening, strain softening, strain rate hardening, and third invariant dependence. The RHT model adopt three strength surfaces to define the failure surface, elastic limit surface, and residual surface as shown in Figure 3.

The failure surface $Y_f$ is can be expressed as a function of pressure $P$, the lode angle $\theta$, and strain rate $\dot{\varepsilon}$,

$$Y_{\text{fail}}(p*, \theta, \dot{\varepsilon}) = Y(p*) \times R_3(\theta, p*) \times F_{\text{rate}}(\dot{\varepsilon}) \tag{1}$$

where $Y(p*)$ is the compressive meridian and is defined as:

$$Y(p*) = f_c \times \left[ A\left(p* - p*_{\text{spall}} \times F_{\text{rate}}(\dot{\varepsilon})\right) N \right] \tag{2}$$

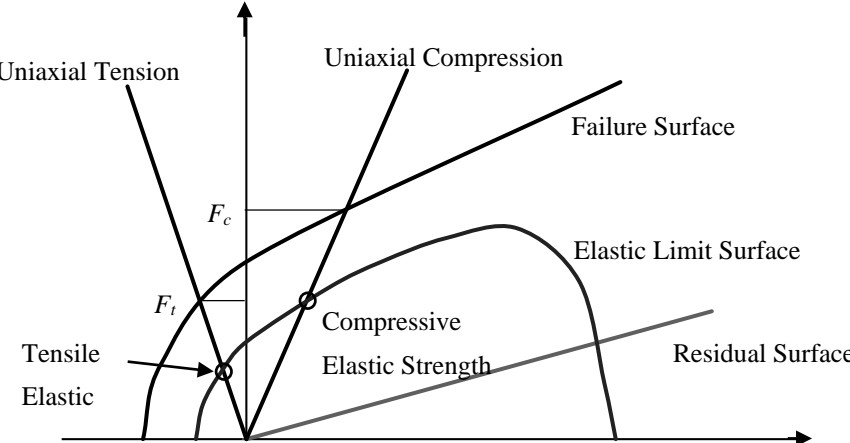

**Figure 3.** Three strength surfaces of RHT model [16].

In which $f_c$ is the material's uniaxial compressive strength, $A$ is the failure surface constant, $p^*$ is the pressure normalized by $f_c$, $p^*_{spall} = p^*(t/f_c)$ ($f_t$ is the material's uniaxial tensile strength), $F_{rate}(\dot{\varepsilon})$ stands for the dynamic increase factor (DIF) function, and $N$ is the failure surface exponent.

$R_3(\theta, p^*)$ is the failure surface ($Y_{fail}$) reduction factor, which is a function of the Lode angle ($\theta$).

The elastic limit surface is scaled from the failure surface,

$$Y_e = Y_{fail} \times F_e \times F_{cap} \qquad (3)$$

where $F_e$ is the ratio of the elastic strength to failure surface strength. $F_{cap}$ controls the elastic deflection stresses under hydrostatic compression and fluctuates in the range of (0,1).

The residual failure surface is defined as:

$$Y_{residual} = f_c \times B \times (p^*/f_c)\, M \qquad (4)$$

where $B$ is the residual failure surface constant, and $M$ is the residual failure surface exponent.

These nominated parameters ($B$ and $M$) control the residual stress and affect the post-failure surfaces, while $D_1$, $D_2$ and $e^{fail}_{min}$, as illustrated in Equations (5)–(7) control the concrete post-softening behavior.

$$D_1 = \int_0^{\varepsilon_p} \left( \frac{\Delta \varepsilon_p}{D_1 \left( P^* - P^*_{spall} \right)^{D_2}} \right) \text{ for } D_1 \left( P^* - P^*_{spall} \right)^{D_2} > e^{fail}_{min} \qquad (5)$$

$$D_2 = \int_0^{\varepsilon_p} \left( \frac{\Delta \varepsilon_p}{e^{fail}_{min}} \right) \text{ for } D_1 \left( P^* - P^*_{spall} \right)^{D_2} < e^{fail}_{min} \qquad (6)$$

$$e^{fail}_{min} = \frac{2G_f}{\sigma_t \times L_{eq}} \qquad (7)$$

$G_f$ is the fracture energy; $\sigma_t$ is the tensile failure stress; and $L_{eq}$ is the characteristic length of the element (the diameter of a sphere with the same size of the 3D element [17]). The adopted modified concrete model in the current work is illustrated in Table 2.

**Table 2.** Constitutive material models adopted in Autodyn.

| Reinforced Concrete (*p-α EOS*, *RHT* Strength, *RHT* Failure) | | | |
|---|---|---|---|
| *EOS* | *p-α* | Compressive strength, $f_c$ (MPa) | 39.5 |
| Reference density (kg/m³) | $2.75 \times 10^3$ | Tensile strength, $ft/fc$ | 0.1 |
| Porous density (kg/m³) | $2.31 \times 10^3$ | Failure surface constant, $A$ | 1.6 |
| Porous sound speed (m/s) | $2.92 \times 10^3$ | Failure surface exponent, $N$ | 0.61 |
| Initial compaction pressure, (GPa) | 0.0233 | Meridian ratio, $Q$ | 0.68 |
| Solid compaction pressure, (GPa) | 6.0 | Brittle to ductile transition | 0.0105 |
| Compaction exponent | 3.0 | Fractured strength constant, $B_c$ | 1.6 |
| Bulk modulus, $A_1$ (GPa) | 35.27 | Fractured strength exponent, $M_c$ | 0.61 |
| Parameter, $A_2$ (GPa) | 39.58 | Damage constant, $D_1$ | 0.04 |
| Parameter, $A_3$ (GPa) | 9.04 | Damage constant, $D_2$ | 1.0 |
| Parameter, $Bo$ | 1.22 | Minimum strain to failure | 0.01 |
| Parameter, $B_1$ | 1.22 | Residual shear modulus fraction | 0.13 |
| Parameter, $T_1$ (GPa) | 35.27 | Principal tensile failure stress (GPa) | 0.015 |
| Parameter, $T_2$ (GPa) | 0.0 | Fracture energy, $G_f$ (J/m²) | 100.0 |
| Reference temp. (K) | 295.0 | Erosion criteria | Geometric strain |
| Specific heat (J/kg K) | 654.0 | Erosion limit | 0.60 |
| Reinforcement steel bars (Linear EOS, Johnson–Cook strength) | | | |
| Reference density (kg/m³) | $7.83 \times 10^3$ | Strain rate constant, $c$ | 0.014 |
| Bulk modulus (GPa) | 159 | Thermal softening exponent, $m$ | 1.03 |
| Shear modulus (GPa) | 81.8 | Indoor temperature, $T_m$ (K) | 300 |
| Yield stress, $A$ (GPa) | 0.792 | Melting temperature, $T_r$ (K) | 1793 |
| Hardening constant, $B$ (GPa) | 0.51 | Ref. strain rate, $\dot{\varepsilon}^0$ | 1 |
| Hardening, exponent, $n$ | 0.26 | | |
| Al 6061-T6 (linear EOS, Johnson–Cook strength, plastic strain failure) | | | |
| Reference density (kg/m³) | $2.7 \times 10^3$ | Strain rate constant, $c$ | 0.01 |
| Bulk modulus (GPa) | | Thermal softening exponent, $m$ | 1 |
| Shear modulus (GPa) | 27.6 | Indoor temperature, $T_m$ (K) | 300 |
| Yield stress, $A$ (GPa) | 0.34 | Melting temperature, $T_r$ (K) | 1220 |
| Hardening constant, $B$ (GPa) | 0.32 | Failure | plastic |
| Hardening, exponent, $n$ | 0.41 | Plastic strain | 0.42 |
| Ref. strain rate, $\dot{\varepsilon}_0$ | 1 | | |
| GFRP composite (ortho EOS, elastic strength) | | | |
| Reference density, (kg/m³) | $1.45 \times 10^3$ | Shear modulus, $G_{12}$ (GPa) | 4.70 |
| Young's modulus, $E_{11}$ (GPa) | 12.10 | Shear modulus, $G_{23}$ (GPa) | 3.10 |
| Young's modulus, $E_{22}$ (GPa) | 6.80 | Shear modulus, $G_{31}$ (GPa) | 4.70 |
| Young's modulus, $E_{33}$ (GPa) | 6.80 | Tensile failure stress, $f_{u11}$ (GPa) | 0.261 |
| Poisson's ratio, $\nu_{12}$ | 0.27 | Tensile failure stress, $f_{u22}$ (GPa) | 0.0261 |
| Poisson's ratio, $\nu_{23}$ | 0.4 | Erosion criteria | Material failure |
| Poisson's ratio, $\nu_{13}$ | 0.27 | | |

Material Model for Reinforcement Steel

Johnson and Cook material model was utilized to simulate the reinforcement steel [18]. This model is ideal for a material under high strain rates, and elevated temperatures. The model of its flow stress is expressed by Equation (8):

$$\sigma_y = \left[ A + B \left( \varepsilon_p^{\text{eq}} \right)^n \right] \left[ 1 + c \ln \left( \frac{\dot{\varepsilon}_p^{\text{eq}}}{\dot{\varepsilon}_0} \right) \right] \left[ 1 - (T^*)^m \right] \tag{8}$$

where $\sigma_y$ is the dynamic stress, $\varepsilon_p^{\text{eq}}$ and $\dot{\varepsilon}_p^{\text{eq}}$ are the equivalent plastic strain and equivalent plastic strain rate, respectively. While $m$ is the thermal softening exponent. $A$, $B$, $n$, $c$, $\dot{\varepsilon}_0$,

and *m* are constants and could be obtained from the flow stress data. $T^*$ is the homologous temperature and it could be calculated from Equation (9).

$$T^* = \frac{T - T_r}{T_m - T_r} \tag{9}$$

*T* is the material temperature; $T_m$ is the material melting temperature, and $T_r$ is the room temperature. To capture the rupture of the reinforced concrete, a failure criterion was adopted on the basis of equivalent plastic strain.

The constants of the steel material developed in the present research were obtained from data of steel 4340 material. The mechanical properties are: bulk modulus, *K* = 159 GPa reference density, $\rho$ = 7.83 g/cm$^3$, reference room temperature, $T_{room}$ = 300 K; specific heat = 477 J/kg *K*; shear modulus, *G* = 81.8 GPa; yield stress, *A* = 792 MPa; hardening constant, *B* = 510 MPa; hardening exponent, *n* = 0.26; strain rate constant, *C* = 0.014; thermal softening exponent, *m* = 1.03; and melting temperature, $T_{melt}$ = 1793 K.

Material Model for Air and TNT

In blast models, the surrounding air and the product of the TNT explosion were supposed to behave like an ideal gas. An ideal gas equation of state (EOS) was used to describe air and was expressed by:

$$P = (\gamma - 1)\rho_g e_0 \tag{10}$$

where *P* is the hydrostatic pressure, $\gamma$ is the ideal gas constant and is 1.4 for air, $\rho_g$ is the density of the air, and $e_0$ is the specific internal energy. The internal energy of air was used as $2.068 \times 10^5$ kJ/kg. This internal energy initialized the air medium to an atmospheric pressure of 101.3 kPa.

The Jones–Wilkins–Lee (JWL) equation of state was used to model high explosive material such as TNT [19], which is in the form of:

$$P = A\left(1 - \frac{\omega}{R_1 V}\right)e^{-R_1 V} + B\left(1 - \frac{\omega}{R_2 V}\right)e^{-R_2 V} + \frac{\omega E}{V} \tag{11}$$

where *A*, *B*, $R_1$, $R_2$, and $\omega$ are empirically derived constants that depend on the type of explosives, *V* is the volume of charge, and *E* is the detonation energy per initial unit volume [20]. TNT's material properties used in the present study *A*, *B*, $R_1$, $R_2$, and $\omega$ are 373.75 GPa, 3.747 GPa, 4.15, 0.9, and 0.35, respectively.

### 2.1.2. Numerical Model

For blast loading, the shock wave interacts with the structure through fluid structure interaction (FSI). Explicit finite element programs are able to simulate this kind of interaction. In the presented work, all the numerical simulations were conducted by utilizing ANSYS/Autodyn V-19. R2. It is a piece of engineering hydrocode software designed to solve nonlinear dynamic problems for instance blast impact exploiting Eulerian, Lagrangian, and Arbitrary Lagrange–Euler (ALE) solvers [17].

The computational cost of the 3D model for explosion simulations is quite expensive (consuming time and needs high computer storage). To overcome this problem with maintaining precise results, two techniques were adopted in this study. The first was taking the advantage of symmetry so a quarter of the model has been generated. The second was using remapping technique, which is an adequate approach to overcome the full 3D structure's meshing problem. This technique allows a 2D model with fine mesh to be mapped into a 3D model with a coarser mesh. The remapping is usually done through a 2D axisymmetric model with a 1 mm element size that was created to simulate the detonation of the explosive charge, as shown in Figure 4. The 2D model was run until the shock vector just before reaching the concrete panel. Next, a remap file has been created and then imported to fill the 3D Eulerian domain (air block) as an initial condition, as shown in Figure 5.

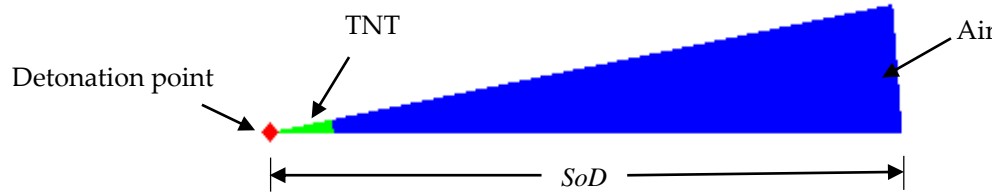

**Figure 4.** Geometry of 1D wedge filled with TNT and air with axial symmetry.

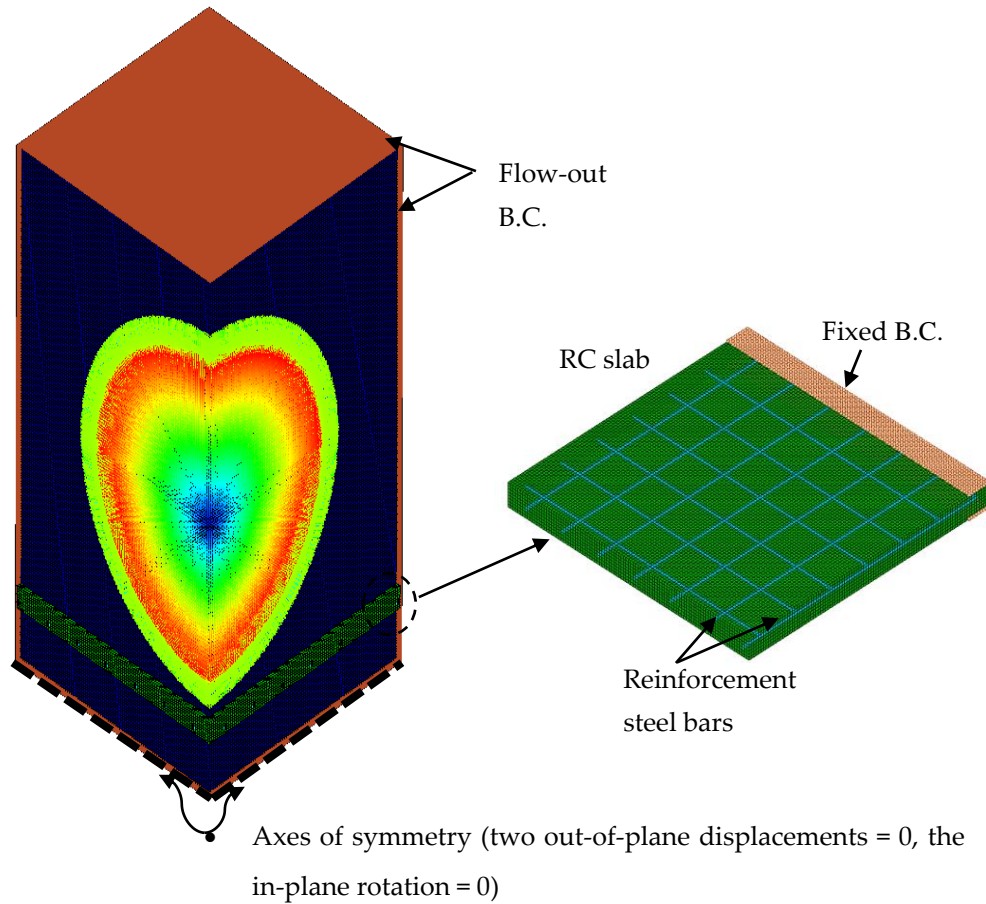

**Figure 5.** 3D FE model for RC panels under blast impact.

The Lagrange solver was employed to simulate solid continua (concrete panel) as the mesh move with the material distortion, Euler solver was adopted to model the gas flow resulted from an explosion as Euler supposes that the material can flow through fixed grid, while beam element was utilized to simulate the reinforcement steel bars. The FE model consists of air domain in which the explosion was initiated. The boundary condition of the Euler sub-grid (air domain) was set as a flow-out boundary at the four faces of the air block as illustrated in Figure 5. In the area supporting the slab, a fixed boundary condition was applied by restraining all the translational degrees of freedom for the nodes located on that edge. A reinforcement bond is considered between the steel bars and the concrete. The element size was selected to be 5 mm to attain consistent results. This size was selected, relying on an executed mesh sensitivity study. The erosion technique was implemented to model the severe damage that occurred to the panels such as spalling that might occurs at the bottom surface of the slabs. An instantaneous geometric strain of 0.6 was adopted in this study.

## 2.2. Results and Discussion

### 2.2.1. Midspan Deflection of the Panels

The displacement-time histories at the mid span of the concrete panel structures due to blasting load were captured using the proposed finite element models as shown in Figure 6. The performance of the RC panels was investigated under the effect of ignition of several TNT charges at different SoD as stated in Table 1. In each case, the displacement-time history at the mid-span is plotted. Panels exhibit more extensive deformation in conjunction with the increasing TNT charge. Panel A attained 8.74 mm residual deformation obtained from the numerical simulation whereas, the experimental residual displacement was 9 mm with an error of around 3%. Panel B showed 25.4 mm central residual deflection while versus 26 mm based on the experimental result with a difference of 2.3%. In addition, panel C showed 14.8 residual central deflection compared 15 mm resulted from the experiments with a discrepancy of 1.3%. Finally, Panel D demonstrated 33.5 mm residual displacement while it deflected by 35 mm in the field blast test with a difference of 4.2%. The numerical central deflections are less than the experimental deflections, and this difference may be due to the boundary conditions are always idea in the numerical simulation, and also a full bond assumed between the concrete and the rebars, however, these ideal conditions does not exist in the field test.

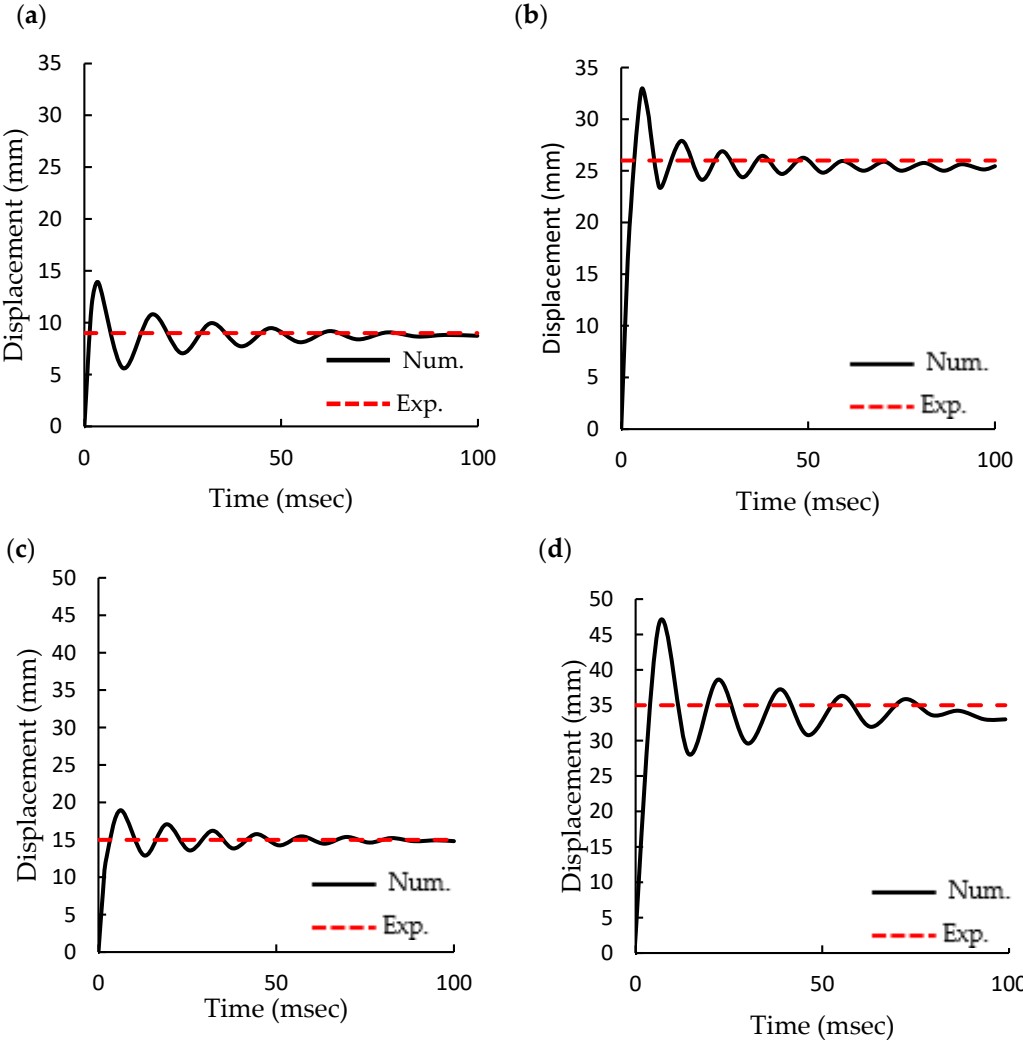

**Figure 6.** Numerical displacement time history and experimental deformations for; (**a**) Panel A, (**b**) Panel B, (**c**) Panel C, (**d**) Panel D.

### 2.2.2. Damage Patterns

Figures 7 and 8 show that the damage occurred at the top and bottom faces of the concrete slabs. The damage level raises with increasing the TNT charge. For comparison purposes, the damage contours obtained from numerical simulations and the damaged experimental panels were displayed. The damaged areas for the top of the panels as the following; those from Panel A under a 0.13 kg TNT charge and with SoD 0.3 m showed no evident damage, except for some minor cracks at the center of the slab surface. For panel B subjected to the detonation of 0.19 kg TNT charge with SoD 0.3 m, several small cracks were also observed in the center area, which resulted from the high pressure of the explosion. Panel C, under the impact of 0.31 kg TNT charge with SoD 0.4 m showed a 3 mm crack through the mid-span of the slab. For panel D, there were circular and radial cracks with a small damaged area in the center of the upper side of the slab, which matches to the test results. In the numerical simulation, only cracks occurred at the top surface of the concrete panels while the bottom surface experienced spalling.

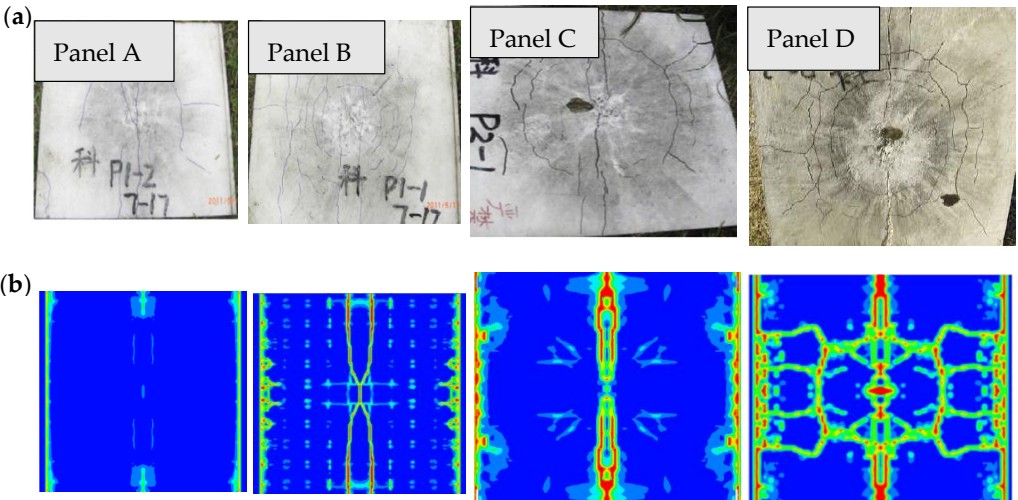

**Figure 7.** The upper surface damage of the RC slabs; (**a**) Experimental examination [14], (**b**) FE simulations.

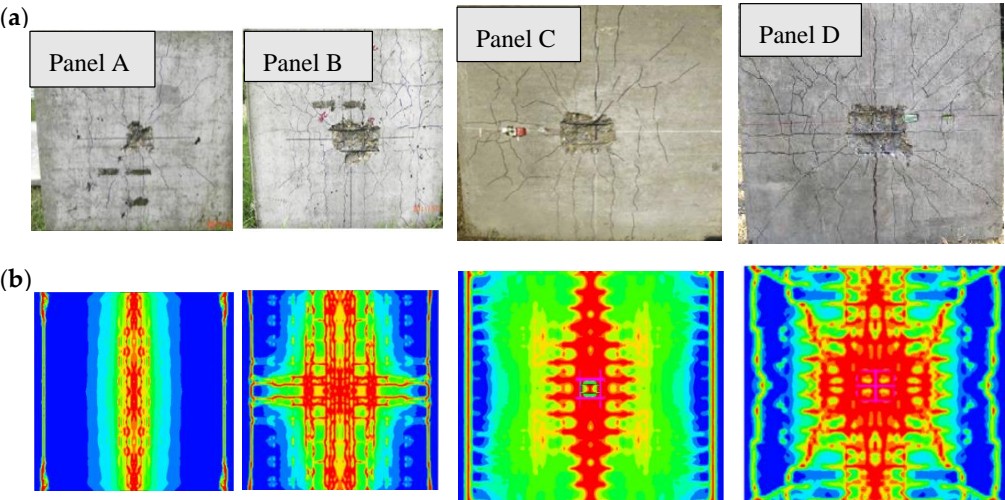

**Figure 8.** The bottom surface damage of the RC slabs; (**a**) Experimental examination [14], (**b**) FE simulations.

The bottom surface of the slab panels suffered from higher damage compared to the upper surface. The bottom damage area and patterns are shown in Figure 8. It illustrates a

comparison between the damaged areas for the panels' bottom surface obtained from the numerical simulation and the field blast test. Panel A attained spallation with a radius of 45 mm lower than the exact value by 10%. For panel B, the numerical simulation presents the radius of the damaged area as 95 mm, which is wider than the experimental damage radius by 11.7%. Similarly, spalling occurred on panel C's bottom surface in the test, as shown in Figure 8. The computed damage area was approximately 100 mm with a 10% discrepancy from the test results, which is in good agreement with the numerical approach. This panel experienced from moderate damage.

Additionally, spalling occurred in panel D bottom surface is illustrated in Figure 8. The computed damage area on the lower surface was approximately 135 mm, which is wider than the experimental by about 12.5%. The slab also showed a severe damage. The numerical radii of the spall area were slightly larger than the experimental test, which might be due to erosion arithmetic and material constants. Nevertheless, these differences are within the limit and reasonable to asses the blast performance of the panels. Table 3 illustrates the experimental and simulation results.

**Table 3.** Experimental test and numerical models results.

| Panel | Mid-Span Deflection (mm) | | Discrepancy (%) | Spall Radius (mm) | | Discrepancy (%) |
|---|---|---|---|---|---|---|
| | Exp. [14] | Numerical | | Exp. [14] | Numerical | |
| Panel A | 9 | 8.74 | 3.0% | 50 | 45 | 10.0% |
| Panel B | 26 | 25.4 | 2.3% | 85 | 95 | 11.7% |
| Panel C | 15 | 14.8 | 1.3% | 90 | 100 | 10.0% |
| Panel D | 35 | 33.5 | 4.2% | 120 | 135 | 12.5% |

Panel D was selected to highlight the competence of the new proposed sacrificial cladding structures for protecting RC panels from blast hazards.

## 3. Proposed Sacrificial Cladding Structure

This section presents a sacrificial cladding structure consisting of a set of thin-walled hybrid structures that should be account for mitigating the significant portion of energy resulted from blast loading. Numerical models had been conducted for three configurations of the hybrid tubes; hybrid single cell tube (H-SCT), hybrid double cell tube (H-DCT), and hybrid quadrable cell tube (H-DCT) as illustrated in Figure 9. It is crucial to determine the behavior and characteristics (energy absorption, deformation pattern, peak crush load, etc.) of the standalone crushable core layer before designing a full-scale cladding structure. The authors themselves have investigated the energy absorption capabilities for metallic and hybrid multi-cell tubes [21]. They all showed a progressive crushing performance and desirable energy absorption capacity compared to solo metallic tubes. The sacrificial layer is designed with r (tube radius) distance in-between tubes, as shown in Figure 9. To allow the incidence of the progressive failure of the core layers. The core layer was covered with a front skin sheet made of aluminum with a 2 mm thickness. In contrast, the core layers presented in this study were fabricated from hybrid tubes wrapped with four uni-directional CFRP sheets. The CFRP layers had ($0°/90°/0°/90°$) layout around the AA6061-T6 tubes with three different configurations as illustrated in Figure 9. The tubes' dimensions were 1.2 mm wall thickness, 60 mm inner diameter, and 120 mm in total length. The inner ribs were done to suit the tubes' inward diameter to obtain the desired multi-cell tubes' configuration. The RC panel's blast performance was numerically investigated utilizing ANSYS/Autodyn under the effect of the blast loads. Panel D expressed the highest level of damage, so it was selected to highlight the effectiveness of the new proposed sacrificial cladding structures for protecting RC panels from blast hazard. Four-noded Belytschko–Tsay shell elements were used to model the front skin plate and the metallic tubes; however, a composite shell was adopted to represent the wrapped composite sheets. The RC panel was modeled as specified before. In the model,

stress-criteria breakable bonded face connection was assumed between the structural elements (front skin, metallic tubes, CFRP tubes, and RC panel). The failure parameters listed in Table 3 were assumed for different breakable contact types. The constitutive models for the adopted materials are listed in Table 3. The whole structure was located in the created air domain and exposed to the blast loading produced by detonating 0.46 kg of TNT located at 0.4 m from the upper surface of the concrete and 0.386 m from the front skin plate.

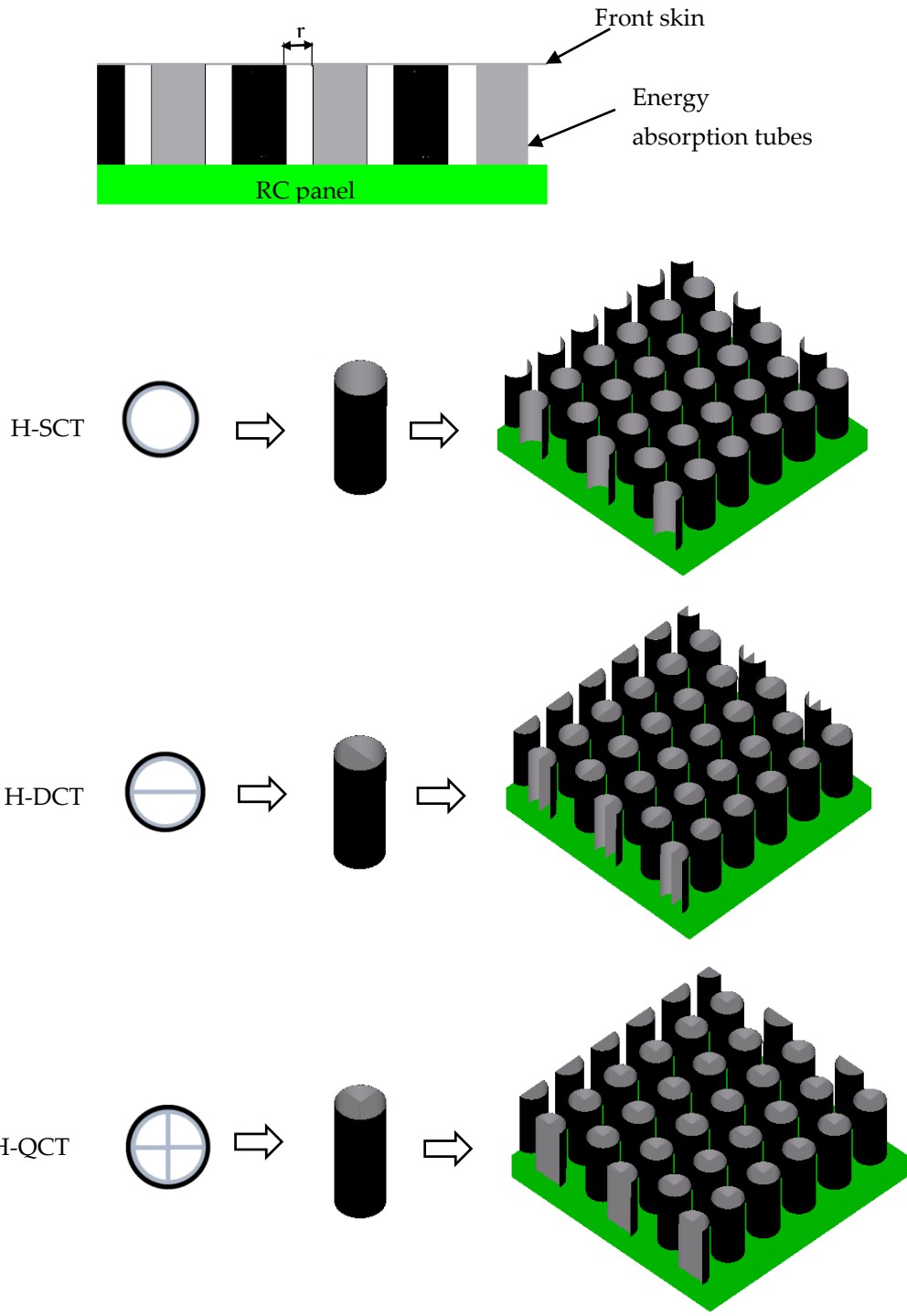

**Figure 9.** Schematic and inner core structures of the proposed sacrificial cladding layers.

## 4. Blast Performance of Protected RC Panels

ANSYS/Autodyn V-19.0. R2 was also adopted to study the dynamic performance of the protected RC panels subjected to the impact of blast loading. The blast wave firstly struck the front skin plate, and fluid-structure interaction (FSI) took place. The front skin face acquired an initial velocity once the shock wave impacted on it and deformed. Then, the skin plate distributed the blast load more consistently on the core layer which was shaped from hybrid multi-cell thin-walled tubes.

The tubes began to mitigate the significant portion of the blast load (pressure) through the AL components' progressive plastic deformation and the wrapped CFRP layers' delamination and fracture. Consequently, the pressure reached the RC panel was attenuated. The entire structure went into oscillation until the kinetic energy was gradually dispersed by stretching and plastic bending has occurred. Finally, the panels underwent residual deformation, as shown in Figure 10. In this study, the plastic deformation and damage patterns were displayed to highlight the effectiveness of applying novel sacrificial cladding structures to the concrete structures. Figure 10 shows that the RC panel shielded with sacrificial layer has attained residual deformations of 12.69, 7.31, and 4.52 mm for H-SCT, H-DCT, and H-QCT cores, respectively. It is concluded from the results that the front skin plate suffered from larger deformations than the RC deformation due to the energy dissipated by the cladding structure cores. Thus, a sacrificial cladding structure is a distinctive approach to protect structures from blast hazards. Additionally, damage patterns were displayed in Figure 11. Applying this technique is very useful as the cladding structure prevents the RC panel from spallation and the bottom of the RC panels had tiny cracks. The level of damage gradually decreased from extreme damage for the bare concrete to just tiny cracks for the RC panel protected by H-QCT sacrificial structure. Table 4 indicates the damage level for the protected and unprotected RC panels.

**Table 4.** Panels' damage levels.

| Panel | Damage Level |
|---|---|
| Un-protected RC panel | Sever damage |
| H-SCT protected panels | Moderate damage |
| H-DCT protected panels | Low damage |
| H-QCT protected panels | Low damage |

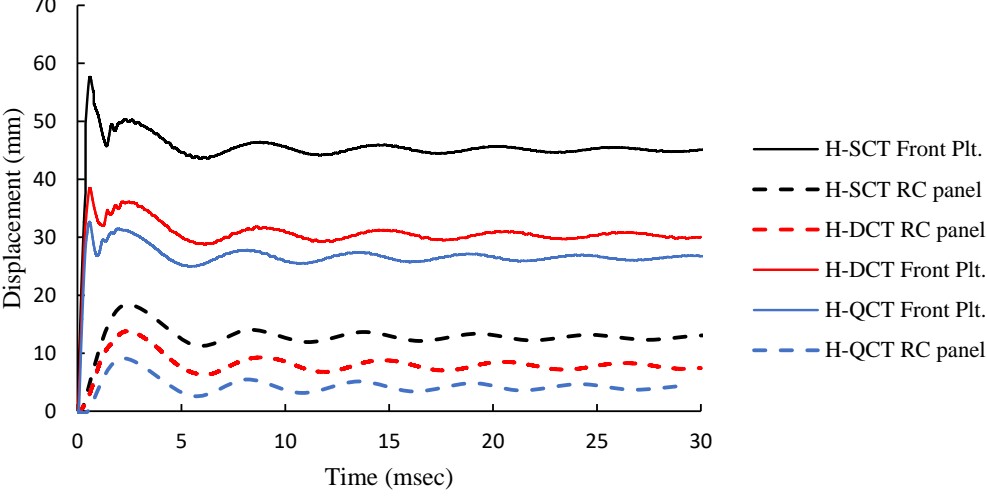

**Figure 10.** Mid-span displacement time history for Panels with different cladding structures (H-SCT, H-DCT, and H-QCT cladding structures).

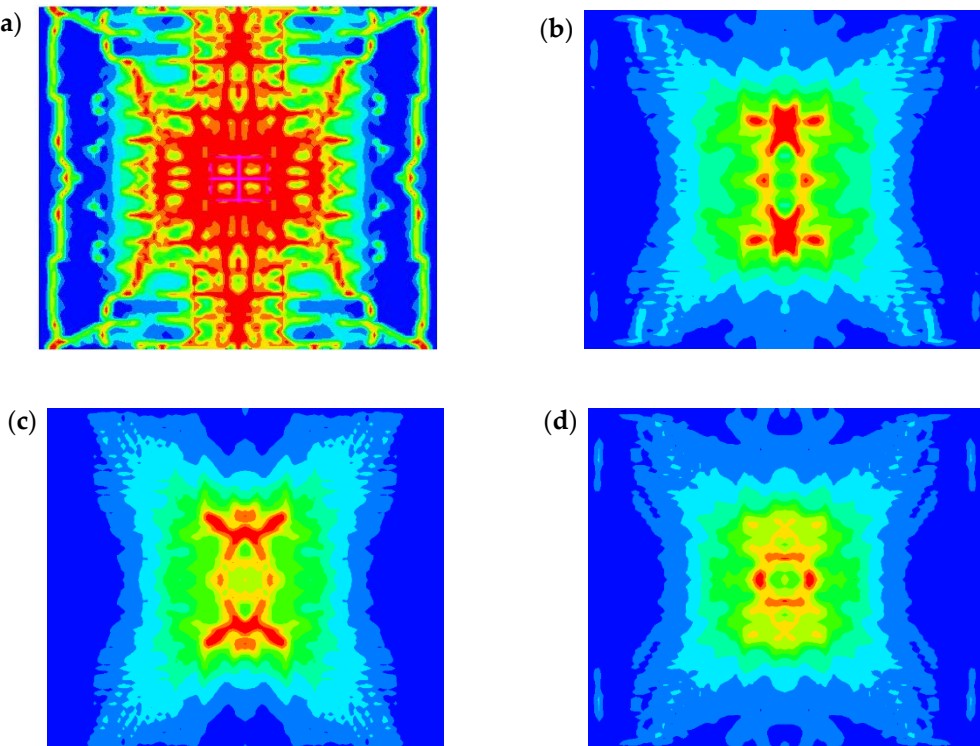

**Figure 11.** Damage patterns on the lower surface of the RC slabs: (**a**) un-protected panel, (**b**) Panel with H-SCT cladding structure, (**c**) Panel with H-DCT cladding structure, (**d**) Panel with H-QCT cladding structure.

Additionally, the presented sacrificial cladding layers were compared with previous implemented sacrificial cladding structures in order to highlight the effectiveness of the cladding structures presented in this study. The improvement for the final maximum deflections were used to assess the difference between other systems and the proposed technique as displayed in Table 5.

**Table 5.** Improvement percent for different cladding systems.

| Reference | Cladding Structure | Improvement |
|---|---|---|
| Mazek et al. [11] | rigid polyurethane foam (RFP) cladding layers | 45.0% |
| | Aluminum foam (ALF) cladding layers | 70.0% |
| Codina et al. [12] | Steel jacketing | 57.4% |
| | Reinforced resin panels | 66.0% |
| Current work | H-SCT cored layers | 62.0% |
| | H-DCT cored layers | 78.0% |
| | H-QCT cored layers | 87.0% |

## 5. Effect of Front Plate Thickness Variation on the Blast Behavior of the Sacrificial Layers

The front skin plate has a crucial contribution to the sacrificial cladding structures' working mechanism as it is responsible for distributing the impact load on the core layer. Thus, this study was extended to investigate the influence of varying the front plate thickness on the presented cladding structures' blast behavior. Four different front plate thicknesses of 2, 4, 6, and 8 mm were applied. The peak deformation of the skin plate and the relative dissipated energy by each core configuration were numerically attained and displayed in Figures 12–14. The dissipated energy by the core layer came out from the variance between the total energy (ET) and the initial energy (EI). Figure 12 demonstrates

the peak deflections for the four different scenarios of the H-SCT cores and the energy dissipated by them. The results illustrates that the contribution of the front skin sheet was significant for the peak deflections of the front plate. Overall, the peak deflection has decreased by increasing the front skin thickness (reduction of about 74.5% by increasing the thickness from 2 mm to 8 mm). While it had less effect on the RC panels' peak deflection as it was decreased by 40%. For the energy dissipated by the core layer, increasing the front skin thickness had a negative impact on it as the energy was dissipated by the core layer reduced by 90.8%. For H-DCT cores, the results followed the same trend as peak deflection, where the energy of the front plate has decreased by 58%, and the peak deflection of the RC panel has also decreased by 33.1%. In addition, the energy dissipated by the tubes have reduced by 92.8%, as shown in Figure 13. Figure 14 displays the results obtained for the H-QCT cores. The peak deflection of the front plate has decreased by 53.5% and the peak deflection of the RC panel decreased by 64.8%. Additionally, the energy dissipated by the tubes decreased by 85.8%. The results indicate that when using a 2 mm thick front plate shell, the energy dissipated attains the highest value and then decreases until it reaches the minimum when the front plate thickness is 8 mm. It could be concluded the energy dissipation decreases when the thickness of the front panel increases. In conclusion, sacrificial cladding structures with a thin front skin plate could improve the energy absorption capabilities of the structure. However, under intense blast loading, the thinner front face rupture damage may occur.

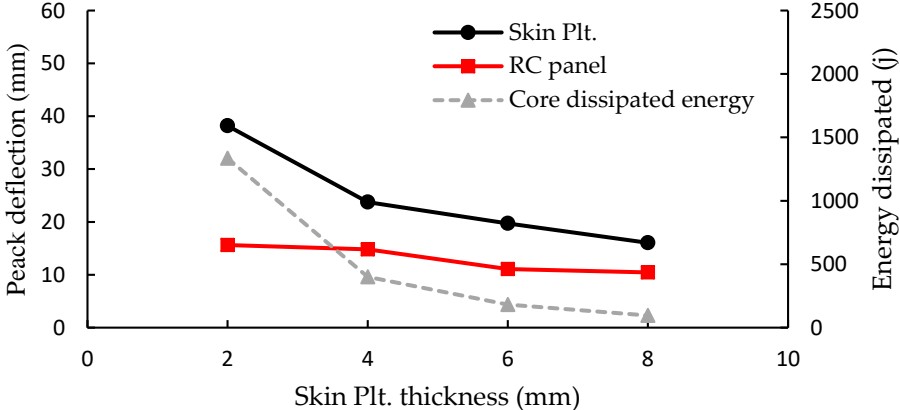

**Figure 12.** Peak deflections and energy dissipated with skin plate thickness variation for RC Panel with H-SCT cladding structure.

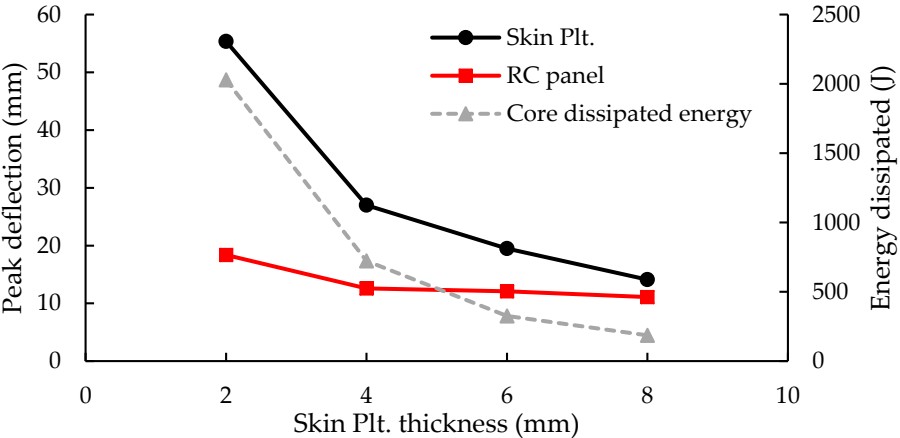

**Figure 13.** Peak deflections and energy dissipated with skin plate thickness variation for RC Panel with H-DCT cladding structure.

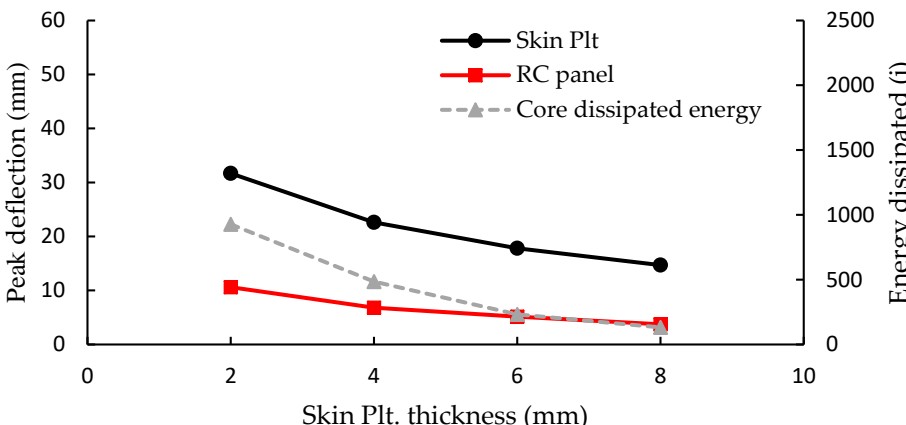

**Figure 14.** Peak deflections and energy dissipated with skin plate thickness variation for RC Panel with H-DCT cladding structure.

## 6. Conclusions

This study's main contribution is to present a new sacrificial structure with hybrid multi-cell tubes as an effective energy absorber component. The sacrificial structure has been proposed for protected and un-protected concrete panels. Numerical simulations of four blast tests were executed to verify the damage patterns of RC panels subjected to a close-in blast loading under various TNT charges. Nonlinear 3D explicit FE models, consisting of air domain, explosive, and RC slab (reinforcing steel bars inside plain concrete), were generated to validate the experimental results obtained by Wang et al. [14]. The advanced models of concrete material and reinforcing bars, taking into account the effects of the high strain rate and proper coupling interface between the Euler and Lagrange elements (the explosion domain and the structure), were exploited to simulate the RC slab's dynamic behavior. The erosion technique was also used to model the damage process.

A good agreement was accomplished through numerical models to predict the deformation/damage patterns of the blast field-tested panels, attaining a maximum deviation of 4.2% and 12.5% for the mid-span deflection and spall radius, respectively. The FE models were employed to study the behavior of the proposed protective structures under the same conditions of the experimental tests to highlight the influence of using sacrificial cladding structures for improving the blast performance of the RC panels. Three different core configurations (H-SCT, H-DCT, and H-QCT) were adopted as core layers for the cladding structures. The dynamic performance of the protected RC panels was studied under in-close blast load by adopting nonlinear explicit finite element models. Results revealed that the cladding structure attained a desired protection for the RC panel as the residual deformations decreased by 62%, 78%, and 87% for H-SCT, H-DCT, and H-QCT cores, respectively, compared to the unprotected panel, which indicates that a large portion of the blast energy was mitigated. Furthermore, the damage pattern for the shielded panels improved from severe damage and spallation to just minor cracks on the back face of the RC panel. A parametric study was performed to investigate the effect of skin plate thickness on the blast performance of the cladding structure. Slab deflection decreased as the front skin plate thickness and stiffness increased. However, it had a negative impact on the deformation of the core layer and its energy dissipation. Increasing the thickness of the skin plate, a larger portion of energy was dissipated by the skin plate, so the core layer did not engage with its full capacity. To conclude, the novel proposed sacrificial structures have shown superior blast shielding for the RC structures.

**Author Contributions:** All authors have contributed to the paper as follows; Conceptualization, M.A., A.I. and S.J.J.; methodology, A.I. and M.A.; software, M.A.; validation M.A.; formal analysis, A.I. and M.A.; investigation, A.I. and M.A.; resources, A.I. and M.A.; data curation, A.I. and M.A.; writing—A.I., S.J.J. and M.A.; writing—review and editing, A.I., S.J.J. and M.A.; supervision, A.I. All authors have read and agreed to the published version of the manuscript.

**Funding:** This research received no external funding and the APC was provided by the University of Idaho Open Access Publishing Fund.

**Institutional Review Board Statement:** Not applicable.

**Informed Consent Statement:** Not applicable.

**Data Availability Statement:** The data presented in this study are all included in this article.

**Conflicts of Interest:** The authors declare no conflict of interest.

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
