# Peer review of "Improving Blast Performance of Reinforced Concrete Panels Using Sacrificial Cladding with Hybrid-Multi Cell Tubes"

_2673-3951, doi:10.3390/modelling2010008_

Round 1

Reviewer 1 Report

The manuscript entitled "Improving Blast Performance of Reinforced Concrete Panels using Sacrificial Cladding with Hybrid-Multi Cell Tubes” constructed a sacrificial cladding structure from multicellular hybrid tubes 11 to protect the prominent bearing members of civil engineering structures from the blast hazard. Eulerian-Lagrangian coupled analyses were conducted using the explicit finite element program (Autodyn/ANSYS) to investigate the proposed problem. The numerical models’ accuracy was validated with available blast testing data reported in the literature.

The manuscript is good in quality and well written. However, this reviewer recommends major editing before acceptance.

Technical comments:

  • Why did the authors put Table 3 on page 13?
  • Line 209: "the structure" is repeated more than once.
  • Line 215: What do you mean by expensive? Do you mean consuming time or needs high computer storage?
  • Lines 217-218: What are the used boundary conditions? The authors need to show these boundaries in Fig. 5 at the axes of symmetry.
  • Lines 234-235: What is the type of connection between the steel bars at intersections (the reinforcement mesh)?
  • Lines 250-256: The information provided in the text is not matching with Fig. 6. The authors should clarify this. Just to make sure, do the vertical axes in Fig. 6 represent the displacement which is different than the residual deflections? If so, clarifications should be added.
  • Lines 257-258: Do you think the boundary conditions are the only reason for that. What about the full bond assumed between the concrete and steel rebars? Also, the constitutive models of materials.
  • Why are the experimental displacements showing constant values in Fig. 6? I think due to the effect of the explosion waves, we should expect variable displacement with time.
  • Figure 9: I am not sure if the authors can modify this figure to highlight what are the differences between the three configurations.
  • Lines 323-324: What are the features of this element? Like degrees of freedom. The same also for the composite shell element.
  • Figure 10: This reviewer highly recommends combining the behavior of all the concrete panels with different protection schemes to show the effect of different cladding structures.
  • Figure 10: Panel with H-QCT cladding structure is not shown in the figure????
  • Figures 12-14: I believe the horizontal axes in these figures should be the plate thickness, not the time. This mistake must be corrected.

Author Response

The authors thank the reviewer for the valuable comments. Please find attached the response to the comments and the responses are highlighted in the manuscript. 

Reviewer 2 Report

General Comment

The manuscript presents a numerical study on the blast performance of reinforced concrete (RC) panels using a novel sacrificial cladding with hybrid-multi cell tubes. Three core topologies were tested for the hybrid-multi cell tubes (H-SCT, H-DCT and HQCT). The analysis is performed by using nonlinear finite element method. In a first stage, the experimental results reported in the literature and related with a series of unprotected RC slabs under blast test are used to validate the numerical model. Then, the numerical model is used to simulate one of the RC panels with sacrificial cladding incorporating the different solutions for the hybrid-multi cell tubes. The performance is measured in terms of the mid-span deflection and damage pattern. In addition, a parametric study was performed in which the thickness of the front plate of the cladding structure varied. The results were compared in terms of the peak deflection and energy dissipation.

The manuscript describes the materials, the numerical model and the simulation procedure. The results are presented and discussed in light of the studied variables. From the obtained results, it is concluded that the proposed sacrificial cladding with hybrid-multi cell tubes can be considered a novel and alternative solution to attain the desired protection of RC panels under blast loading.

The topic of the manuscript is an important one since novel and economical sacrificial protective solutions against blast threats are of great need. The results of the presented study could be useful for further studies and also for industry.

I made some comments in order to improve the manuscript. The authors should take the comments into account and revise their manuscript.

Specific Comment 1

The article must be entirely reviewed to improve the check spelling of several sentences and also correct typos.

Specific Comment 2

Introduction + references

The number of references and the literature review are somewhat poor and must be updated. Several recent studies on the topic exist in the literature. See for instance the very good literature review of the following very recent study: https://www.mdpi.com/1996-1073/14/1/214

Specific Comment 3

Section 2.2.1 + Figure 6 + Table 2

In the text, the reported values for the midspan deflection for each panel does not match with the name of the panels in the graphs presented in Figure 6. Please check and correct. Same for Table 2.

Specific Comment 4

In section 4, before Figure 10, it is stated that “Table 3 indicates the damage level for the protected and unprotected RC panels.” This table seems to be missing, since the presented Table 3 presents the constitutive material models adopted in the software.

Specific Comment 5

Figure 10

One graph is missing in Figure 10 since in the caption 4 panels are referred (a, b, c and d) and only 3 graphs exist (a, b and c).

Specific Comment 6

Figures 12 to 14

The captions must me reviewed (for instance, the captions of Fig. 13 and 14 are the same). In addition, information about the thicknesses of the front plate should be included in the graphs.

Specific Comment 7

A brief comparative analysis and discussion with the results of related previous studies is missing. This is important to improve the article and to compare the performance of the proposed solution with other ones using also sacrificial cladding structures.

Author Response

(The authors gave the same response as above.)

Round 2

Reviewer 1 Report

The authors addressed most of the comments that have been highlighted by the reviewer. however, there are two points that the authors misunderstood:

1- Point (4): What are the used boundary conditions? The authors need to show these boundaries in Fig. 5 at the axes of symmetry. I asked for the boundary conditions at the axes of symmetry, not boundaries used to represent the fixed support. The authors mentioned that the quarter panel was modeled to be less expensive. So, show the boundary conditions at the axes of symmetries.

2- Point (11): Figure 10: This reviewer highly recommends combining the behavior of all the concrete panels with different protection schemes to show the effect of different cladding structures. I did not ask to compare your results with any other results. You provided three different strengthening schemes. So, combine the Mid-span displacement time history of the RC panel for all the cladding structures that you used in one figure to be able to see the difference in behavior.

Author Response

The authors thank the reviewer for the comments. The response is attached. 

Reviewer 2 Report

I´m generally satisfied with the authors’ replies to my earlier comments and I also consider that most of my suggestions and concerns have been explained and / or considered by the authors to improve the manuscript. 

Author Response

(The authors gave the same response as above.)
